# The Therapeutic Potential of Orange Juice in Cardiac Remodeling: A Metabolomics Approach

**DOI:** 10.3390/metabo15030198

**Published:** 2025-03-13

**Authors:** Priscila Portugal dos Santos, Anderson Seiji Soares Fujimori, Bertha Furlan Polegato, Marina Politi Okoshi

**Affiliations:** Internal Medicine Department, Botucatu Medical School, Sao Paulo State University (UNESP), Botucatu 18618-687, Brazil; seiji.fujimori@unesp.br (A.S.S.F.); bertha.polegato@unesp.br (B.F.P.); marina.okoshi@unesp.br (M.P.O.)

**Keywords:** orange juice, energy metabolism, gut microbiota, cardiac remodeling, cardiovascular diseases

## Abstract

Cardiovascular diseases are a leading cause of death worldwide, and the process of cardiac remodeling lies at the core of most of these diseases. Sustained cardiac remodeling almost unavoidably ends in progressive muscle dysfunction, heart failure, and ultimately death. Therefore, in order to attenuate cardiac remodeling and reduce mortality, different therapies have been used, but it is important to identify adjuvant factors that can help to modulate this process. One of these factors is the inclusion of affordable foods in the diet with potential cardioprotective properties. Orange juice intake has been associated with several beneficial metabolic changes, which may influence cardiac remodeling induced by cardiovascular diseases. Current opinion highlights how the metabolites and metabolic pathways modulated by orange juice consumption could potentially attenuate cardiac remodeling. It was observed that orange juice intake significantly modulates phospholipids, energy metabolism, endocannabinoid signaling, amino acids, and gut microbiota diversity, improving insulin resistance, dyslipidemia, and metabolic syndrome. Specifically, modulation of phosphatidylethanolamine (PE) metabolism and activation of PPARα and PPARγ receptors, associated with improved energy metabolism, mitochondrial function, and oxidative stress, showed protective effects on the heart. Furthermore, orange juice intake positively impacted gut microbiota diversity and led to an increase in beneficial bacterial populations, correlated with improved metabolic syndrome. These findings suggest that orange juice may act as a metabolic modulator, with potential therapeutic implications for cardiac remodeling associated with cardiovascular diseases.

## 1. Introduction

Cardiovascular disease (CVD) is a leading cause of mortality globally, responsible for a significant number of deaths and disabilities. In 2021, CVD accounted for 20.5 million deaths, comprising approximately one-third of all global deaths [1]. CVD can be prevented by addressing behavioral risk factors, such as diet. The process of cardiac remodeling lies at the core of most cardiovascular diseases. Cardiac adaptation to pressure or volume overload, which typically occurs in several CVDs, is associated with cellular and molecular alterations in cardiomyocytes and the interstitial matrix, which leads to anatomic and functional remodeling of the heart. The alterations include myocyte hypertrophy, interstitial fibrosis, increased oxidative stress, inflammation and apoptosis, and changed energy metabolism. Although initially adaptive, the sustained cardiac hypertrophic remodeling results in progressive myocyte dysfunction, heart failure, and ultimately death [2].

Despite treatment, CVDs have a high mortality and disability rates. Therefore, it is important to identify adjuvant factors that can modulate cardiac remodeling [3]. One of these is the inclusion of affordable foods with potential cardioprotective properties in the diet [4,5,6,7,8]. In accordance, the effects of orange juice and its compounds have been evaluated in clinical and experimental studies. Orange juice consumption can improve energy metabolism, antioxidant capacity, anti-inflammatory properties, insulin resistance, metabolic syndrome, and cardiac function, and it can reduce blood pressure and plasma lipides [6,7,9,10,11,12,13,14,15]. Moreover, orange juice positively modulated composition and metabolic activity of the intestinal microbiota [10,16].

In patients with a high cardiovascular risk and in cardiac injury models, such as doxorubicin-induced cardiotoxicity and myocardial infarction, orange juice consumption has been shown to improve endothelial function and left ventricular function, attenuate cardiac oxidative stress and inflammation, and modulate energy metabolism by providing more substrates for energy production [4,5,6,7,8].

The mechanisms by which orange juice leads to these changes are not yet fully understood. Orange juice contains a collection of bioactive compounds, antioxidants, and anti-inflammatory agents, including flavonoids (hesperidin and naringenin), carotenoids (xanthophylls, cryptoxanthins, carotenes), and vitamin C, in addition to other beneficial phytochemicals that have a significant protective effect against certain diseases [17].

Studies using a metabolomics approach have suggested some metabolites and metabolic pathways that may be involved in the beneficial effects of orange juice [6,16,18,19,20]. However, only a few studies evaluated the effects of orange juice consumption on metabolomic analysis in situations of high cardiovascular risk. Current opinion highlights that metabolites and metabolic pathways modulated by orange juice consumption can potentially attenuate cardiac remodeling. Nonetheless, further studies are needed to evaluate the metabolome in CVDs to confirm modulation of metabolic pathways by orange juice.

## 2. Orange Juice Consumption and the Metabolomics Approach

A metabolomics-based study in healthy volunteers evaluated the effect of two weeks of orange juice consumption and showed a decrease in medium- and long-chain acyl-carnitines and an increase in short-chain acyl-carnitines (Table 1) in blood levels, which affects liver fatty acid β-oxidation, including both mitochondrial and peroxisomal β-oxidation [19].

Another study, conducted in overweight and obese adults, found that orange juice consumption decreased oxidative stress, improved anti-inflammatory properties, and modulated gut microbiota metabolism. Increased serum levels of ferulic acid 4-sulfate and dihydroferulic acid were observed. Both compounds have been described as flavanone-derived metabolites synthesized by colonic microbiota [18]. Serum levels of hydroxyoctadecadienoic acid (9-HODE + 13-HODE) and dihydroxyoctadecanoic acid (12,13-DiHOME and 9,10-DiHOME) decreased, while levels of 12-hydroxyeicosatetraenoic acid (12-HETE) increased (Table 1). The authors explained that 9-HODE + 13-HODE is a biomarker of oxidative stress derived from lipid peroxidation present in atherosclerotic plaques. On the other hand, 9,10-DiHOME and 12,13-DiHOME are linoleic acid-derived molecules produced by neutrophils and macrophages, which have toxic effects when accumulated in cells. 12-HETE regulates vasoconstriction, counteracts inflammation and tissue damage, and contributes to platelets function. Moreover, 12-HETE can mediate substrate regulation of the glucose transport mechanism during hyperglycemia. Thus, all these effects may explain the antioxidant and anti-inflammatory properties attributed to orange juice [18].

An experimental study in healthy rats showed that orange juice consumption improved alpha diversity and decreased the gut microbiota Firmicutes/Bacteroidota ratio (F/B ratio), preventing insulin resistance [16].

A previous study conducted by our research group showed that ingestion of Pera and Moro orange juice induces changes in plasma metabolites related to the regulation of extracellular matrix, inflammation, oxidative stress, and membrane integrity in healthy rats [20]. The ingestion of Pera orange juice was associated with serum changes in the metabolites N-docosahexaenoyl-phenylalanine, diglyceride (DG, 20:4/24:1), and phosphatidylethanolamine (PE, O-20:0/16:0) (Table 1). On the other hand, the ingestion of Moro orange juice was associated with serum changes in the following metabolites: casegravol isovalerate, abscisic alcohol 11-glucoside, torvoside C, N-formylmaleamic acid, N2-acetyl-L-ornithine, and cyclic phosphatidic acid (CPA, 18:2) (Table 1) [20]. A more in-depth review of the metabolic pathways related to these compounds reveals that they are mainly associated with phospholipid metabolism, the cannabinoid system, and microbiota metabolism.

N-docosahexaenoyl phenylalanine belongs to the class of compounds known as N-acylamides. N-acylamides are one of the main groups of simple lipids with structures consisting of a fatty acid (acyl group) attached to a simple amine by an amide bond. These compounds fall into several categories, e.g., acylamides conjugated with amino acids, such as N-docosahexaenoyl phenylalanine [21].

Other compounds altered in plasma by the ingestion of Pera orange juice in the study by Fujimori et al. [20] were diglyceride (DG, 20:4/24:1) and phosphatidylethanolamine (PE, O-20:0/16:0). Both compounds participate in the synthesis pathways of the main phospholipids and triacylglycerols in eukaryotes. PE is a glycerophospholipid present in most cell membranes. One of the synthesis pathways for glycerophospholipids, including PE, is the phosphorylation of diacylglycerol, such as diglyceride (DG, 20:4/24:1). A cytidine diphosphate (CDP) is coupled to diacylglycerol, forming the activated phosphatidic acid CDP-diacylglycerol; then, condensation with ethanolamine occurs to form phosphatidylethanolamine [22,23]. The synthesis of ethanolamine from CDP-diacylglycerol is a major PE-producing pathway in eukaryotes [22].

Regarding the ingestion of Moro orange juice, three compounds found to be altered by Fujimori et al. [20] are not intermediate metabolites of mammalian metabolic pathways but are possibly derived from oranges. They are casegravol isovalerate, abscisic alcohol 11-glucoside, and torvoside C.

Casegravol isovalerate belongs to the class of organic compounds known as coumarins and its derivatives. Casegravol isovalerate has been detected in citrus plants [24], including some orange species. Few articles have been published on casegravol isovalerate, and there are no studies showing an association of this compound with cardiovascular changes. Detsi et al. [25] showed that coumarin, a naturally plant-derived or synthetically obtained substance, presents several biological activities and an extensive therapeutic profile. Coumarin derivatives are referred to as receptor modulators with a potential for treating cardiovascular diseases.

Abscisic alcohol 11-glucoside belongs to the class of organic compounds known as terpene glycosides. It is present in some plants and orange species [26]. There are no studies showing its association with cardiovascular changes. However, other glycosidic terpenes have presented antiatherosclerotic effects [27]. Additionally, glycosidic terpenes protect against isoproterenol-induced cardiomyopathy, potentially by improving cardiac energy metabolism and inhibiting cardiomyocyte apoptosis [28].

The other altered compound, torvoside C, belongs to the class of organic compounds known as steroidal saponins. In a report by Yahara et al. [29], torvoside C was identified in the aerial part of *Solanum torvum*. No studies showed the presence of this compound in oranges or its association with cardiovascular disease. However, the fruits of *Solanum torvum* are commonly used in traditional medicine for their antihypertensive, antioxidant, and anti-platelet aggregation activities [30]. Therefore, this compound has the potential to modulate cardiac remodeling, but further studies are needed.

Ingestion of Moro orange juice also appears to be associated with changes in microbiota metabolism. The compounds N-formylmaleamic acid and N2-acetyl-L-ornithine, which were altered in the study by Fujimori et al. [20], are metabolites of oral and intestinal microflora. Studies suggest that the presence of N-formylmaleamic acid may result from the metabolism of vitamin B3 (which is present in *Citrus sinensis* L. Osbeck) by the microbiota [31,32,33]. N-formylmaleamic acid is a metabolite of nicotinamide (Nam) catabolism in several bacteria and some yeasts [27,28,29]. Nam degradation in these organisms is initiated by deamination to form nicotinic acid (NA) [34]. Nam and NA are two forms of vitamin B3 and are precursors in the biosynthesis of the coenzyme nicotinamide adenine dinucleotide (NAD) [23]. In mammals, the degradation pathway of Nam does not form N-formylmaleamic acid [31,35] or NA [36]. Therefore, Nam deamination by oral and intestinal microflora must occur before NA use as a precursor in NAD synthesis [36].

N2-acetyl-L-ornithine is a metabolite of the de novo biosynthetic pathway for ornithine (and, therefore, arginine) in bacteria. The precursor metabolite is the amino acid glutamate. Hydrolysis of the acetyl group from N2-acetyl-L-ornithine forms ornithine, which, through the urea cycle, is converted into arginine [23]. In mammals, the pathways for producing ornithine and arginine are somewhat different. Ornithine can also be synthesized from glutamate by transamination, but the spontaneous cyclization of glutamate semialdehyde, which is directed to the proline synthesis pathway, prevents a sufficient supply of this intermediate for ornithine synthesis. Therefore, arginine synthesis in mammals does not occur through the de novo biosynthetic pathway for ornithine, and the metabolite N2-acetyl-L-ornithine is not formed. Arginine is synthesized through reactions in the urea cycle [23]. Although N2-acetyl-L-ornithine is not a compound of mammalian metabolism, studies investigating plasma metabolomics have shown the presence of this compound in rats [37,38]. Some authors suggest that the changes in ornithine/arginine metabolism observed in humans, rats, and mice may be associated with gut microflora metabolism [39,40,41].

The last compound associated with the ingestion of Moro orange juice in the study by Fujimori et al. [20] was cyclic phosphatidic acid (CPA; 18:2). CPA is a glycerophospholipid in which there is a cyclic phosphate at the sn-2 and sn-3 positions of the glycerol carbons; this structure is absolutely necessary for its activity. CPAs present different combinations of fatty acids of varying lengths and saturation attached at the C-1 (sn-1), with fatty acids containing 16 and 18 carbons being the most common. CPAs have been detected in a wide range of organisms, including humans, mainly in the brain but also in serum [40]. This compound is a naturally occurring analog of the growth factor-like phospholipid mediator lysophosphatidic acid. CPA affects numerous cellular functions, including antimitogenic regulation of the cell cycle, inhibition of tumor cell invasion and metastasis, and regulation of differentiation and survival of neuronal cells [42]. Additionally, studies have shown that CPA plays an important role in cardiovascular diseases. CPA inhibits arterial wall remodeling and inflammatory mediators’ secretion, preventing atherosclerosis [43].

The effects of orange juice intake were evaluated in individuals with a high cardiovascular risk. The study showed that women and men with pre- or stage 1 hypertension presented changes in several metabolites related to anti-inflammatory and antioxidant actions, lower blood pressure levels, and uremic toxins. Increased serum levels of proline betaine (direct marker of citrus fruit intake) and decreased plasma levels of glycerophosphocholine (GPC), acetate, valine, isoleucine, leucine, and N-acetyl glycoproteins (NAGs) were observed with orange juice intake (Table 1) [6]. NAGs are considered a novel biomarker of systemic inflammation and cardiovascular disease risk.

## 3. Metabolic Pathways Modulated by Orange Juice Intake and Potential Attenuation of Cardiac Remodeling

Studies on orange juice intake and metabolomic evaluation have shown that the altered metabolites and metabolic pathways are related to phospholipids, energy metabolism, endocannabinoid signaling, amino acids, and microbiota diversity and metabolism.

### 3.1. Phospholipids and Alterations in Energy Metabolism

The metabolomic studies presented above showed that orange juice intake modulates phospholipids and different metabolites related to energy metabolism. In the study by Fujimori et al. [20], orange juice intake induced the presence of PE and CPA. Both compounds are phospholipids. PE and CPA can modulate the expression and activity of peroxisome proliferator-activated receptors (PPARs) and their target genes [44,45]. Indeed, the activity of PPARs can be influenced by a variety of natural compounds, including polyunsaturated fatty acids, eicosanoids, and oxidized lipid components. PPARs detect fatty acid-derived signaling molecules, which suggests their function as general lipid or nutrient sensors [46,47] (Figure 1).

In the study by Tian et al. [44], Eicosapentaenoic acid-enriched phosphatidylethanolamine (EPA-PE) was shown to bind to PPARα/PPARγ and exert agonistic activity regarding the transcription of PPARα and PPARγ. EPA-PE treatment induced Pparα mRNA expression and its target genes, which encode enzymes of fatty acid β-oxidation, including Carnitine Palmitoyltransferase I (CPT1) in hepatocytes. EPA-PE significantly increased PPARα protein levels in the liver of healthy mice and increased PPARα and CPT1 protein levels in the liver of rodents with insulin resistance, which alleviated hepatic steatosis and lipid accumulation. These results indicate that EPA-PE activates hepatic fatty acid β-oxidation. In models with doxorubicin-induced cardiotoxicity, orange juice intake also increased CPT-1 expression in the liver (Figure 1).

Consistent with these results, orange juice intake in healthy volunteers decreased medium- and long-chain acylcarnitines and increased short-chain acylcarnitine blood levels, which affects both mitochondrial and peroxisomal fatty acid β-oxidation [19]. Acylcarnitines are formed by the binding of an acyl group (derived from fatty acids or related compounds) to carnitine via the carnitine acyltransferase I on the outer mitochondrial membrane. After CPT1-induced production of acylcarnitines, the inner mitochondrial membrane transporter carnitine-acylcarnitine translocase transports acylcarnitines into the mitochondrial matrix. Finally, the enzyme CPT2 reconverts acylcarnitines back into free carnitine and acyl-CoAs, which can then be oxidized [23].

β-oxidation is the primary process by which fatty acids are oxidized through the sequential removal of two-carbon units from the acyl chain to generate acyl coenzyme A (acyl-CoA). Long-chain acylcarnitines (C14 to C20) are essential for the transport and β-oxidation of fatty acids in mitochondria and peroxisomes [23]. Therefore, an increase in long-chain acylcarnitines in plasma or tissue generally occurs when fatty acid entry or metabolism within organelles is abnormal, potentially leading to cellular toxicity, oxidative stress, and reduced energy production [48]. A decrease in long-chain acylcarnitines indicates that fatty acids are physiologically entering the mitochondria for oxidation.

In rodents with doxorubicin-induced cardiotoxicity, orange juice intake improved the activity of 3-hydroxyacyl-CoA dehydrogenase, which is a key enzyme in fatty acid β-oxidation [7]. This enzyme is regulated by the mitochondrial concentration of acyl-CoA, which increases by both the entry of fatty acids into the mitochondria, and the availability of nicotinamide adenine dinucleotide (NAD^+^) and flavin adenine dinucleotide (FAD). The availability of NAD^+^ and FAD depends on mitochondrial oxidative phosphorylation, ATP concentration, and oxygen levels [23]. Ribeiro et al. [7] showed that ATP synthase activity was also increased by orange juice intake. Thus, it is possible that orange juice improves fatty acid β-oxidation in the heart.

The improvement of liver β-oxidation of fatty acids can modulate serum lipid profile and reduce dyslipidemia, which has also been observed with orange juice intake [8,11]. Mitochondrial β-oxidation of fatty acids reduces hepatic and serum triglyceride (TG) levels by increasing the use of fatty acids as an energy source [44,49], decreases total cholesterol (TC) and low-density lipoproteins (LDL) by limiting hepatic cholesterol synthesis [44,50,51], and increases high-density lipoproteins (HDL) by removing cholesterol excess [52], thereby reducing the risk of cardiovascular disease (Figure 1).

The improvement of β-oxidation of fatty acids also plays a positive role in insulin resistance and metabolic syndrome. Altered β-oxidation increases reactive oxygen species production and mitochondrial dysfunction, as well as the accumulation of lipid intermediates, which can interfere with insulin signaling and activate pro-inflammatory pathways, leading to insulin resistance and metabolic syndrome [53]. Studies have shown that orange juice intake improves insulin resistance and prevents the development of metabolic syndrome [9,10,11], which are cardiovascular risk factors.

Homeostasis of β-oxidation of fatty acids in myocardium is crucial since the heart primarily uses fatty acids as an energy source. In conditions of cardiac injury, such as ischemia or myocardial hypertrophy, we observed reduced β-oxidation of fatty acids and increased glucose utilization as an energy source. Glucose oxidation requires less oxygen per ATP produced, making it theoretically more efficient under stress conditions [54]. However, restoring or improving β-oxidation may be beneficial, attenuating cardiac remodeling progression.

A decrease in β-oxidation during cardiac injury can lead to the accumulation of lipid intermediates, such as ceramides and diacylglycerols, which trigger lipotoxicity, oxidative stress, and apoptosis—frequent features in cardiac remodeling [54]. Restoring β-oxidation provides sustainable energy, especially in chronic injury states where energy demand may be high. However, the balance between metabolic pathways needs to be carefully adjusted to avoid disadvantages associated with increased oxygen consumption. An improvement in β-oxidation should be considered in specific contexts and accompanied by appropriate clinical support.

EPA-PE also modulates the expression of PPARγ [44,55]. PPARγ governs the expression of genes involved in lipid and glucose metabolism, including lipogenesis, fatty acid transport, glycolysis, and gluconeogenesis, which play a critical role in glycemic control [47,56]. PPARγ is the primary regulatory factor controlling the insulin signaling pathway and overall insulin sensitivity [56]. Treatment with EPA-PE promoted the expression of PPARγ in adipose tissue. Additionally, EPA-PE reduced adipocyte size, lipid droplets, insulin levels, and fasting blood glucose levels in rodents with insulin resistance. Consistent with these results, serum TG and TC levels were significantly reduced by EPA-PE administration [44]. Other studies have shown that PPARγ activation regulates CPT1, decreases circulating blood lipids, and inhibits liver steatosis [57,58]. Thus, PPARγ can regulate adipose tissue remodeling and attenuate insulin resistance and dyslipidemia (Figure 1).

PPARα and PPARγ also modulate inflammation. They are crucial for activation of immune system cells and release of anti-inflammatory cytokines and chemokines [47]. Additionally, they suppress the expression of inflammatory genes [46,47,56]. Cabral et al. [8] showed that orange juice intake increased myocardial PPARγ expression in a doxorubicin-induced cardiotoxicity model. Thus, PPARγ activation may be involved in orange juice-induced improvement in insulin resistance and dyslipidemia while reducing inflammation and preventing metabolic syndrome (Figure 1).

PPARγ also modulates mitochondrial function and structure. Mitochondria are highly abundant in cardiomyocytes, reflecting the heart’s highly oxidative nature and its dependence on these organelles to generate ATP from fatty acid and glucose oxidation, essential for muscle contraction. Thus, mitochondria play a crucial role in regulating cardiac function in both health and disease. PPARγ increases the expression of components of the respiratory chain complex and mitochondrial biogenesis-related genes [59]. Cabral et al. [8] observed that orange juice decreased serum aspartate aminotransferase (AST) concentration in doxorubicin-treated animals (Figure 1). Since mitochondria contain a large amount of AST, the decrease in AST may be a marker of decreased mitochondrial injury [15,60,61].

In summary, the increase in PE induced by orange juice intake may be associated with the activation of PPARα and PPARγ, which regulate energy metabolism, mitochondrial function, oxidative stress, and inflammatory activation. Myocardial modulation of energy metabolism, mitochondrial function, oxidative stress, and inflammation is related to cardiac remodeling, which can lead to cardiac dysfunction (Figure 1).

PE is the second most abundant phospholipid, representing approximately 37% of total phospholipids, in cardiomyocytes [62]. It participates in the respiratory complex IV [22] and in stabilization of the sarcolemmal membranes during ischemia [63]. PE deficiency impairs oxidative phosphorylation and mitochondrial ultrastructure and was associated with NLRP3 inflammasome activation and cardiac remodeling [64,65].

Other phospholipids and fatty acid-derived metabolites found in the plasma of animals that consumed orange juice, such as CPA, 9-HODE, 13-HODE, 12,13-DiHOME, 9,10-DiHOME, and 12-HETE, are also possible PPAR ligands and may modulate the aforementioned processes [46,66,67,68,69] (Figure 1). Studies have shown that these metabolites exhibit antioxidant and anti-inflammatory effects and play a role in glucose and fatty acid metabolism [18,43,70,71,72].

Another metabolite altered by orange juice intake is glycerophosphocholine (GPC), which was decreased in hypertensive subjects [11]. This compound is a precursor of several important metabolites. GPC cleavage forms choline and glycerol-3-phosphate. Choline is further metabolized to phosphatidylcholine, a key player lipid in membrane-mediated cell signaling [73]. Choline is used in acetylcholine synthesis. Kawamura et al. [74] showed that oral ingestion of GPC acutely increased plasma free choline levels by 50%, leading to a concomitant increase in growth hormone concentration and hepatic fat oxidation in young adult male subjects. Glycerol-3-phosphate is a precursor for lipids with central roles in signaling, such as lysophosphatidic acid, phosphatidic acid, CPA, and diacylglycerol. The latter is a precursor for various phospholipids synthesis, including PE [73] (Figure 1).

Other GPC metabolites include lysophosphocholines and platelet-activating factors, lipids that modulate systemic oxidative stress and inflammation [75,76]. GPC degradation under stress and tissue inflammation releases metabolites with antioxidant and anti-inflammatory action, leading to a decrease in its concentration [6,77]. GPC administration preserved mitochondrial complex I function, suppressed activity of intracellular superoxide-generating enzymes, and reduced biochemical signs of oxidative stress and inflammatory activation in a liver ischemia–reperfusion model [78].

### 3.2. Endocannabinoid Signaling

Endocannabinoids are lipid mediators that act as messengers in intercellular communication. They occur naturally in the body and mimic the activity of Δ9-tetrahydrocannabinol, the primary psychoactive ingredient of cannabis. The endocannabinoid system (ECS) is a widespread cell signaling system composed of a large group of these lipid mediators, their primary receptors, and the associated enzymes responsible for their synthesis and degradation. The most well-studied endogenous ligands are the endocannabinoids N-arachidonoylethanolamine/anandamide and 2-arachidonoylglycerol, and the two main metabotropic receptors that respond to them are named cannabinoid receptor type-1 (CB1) and type-2 (CB2). Together, these components of the ECS work harmoniously to maintain homeostasis [79,80].

Endocannabinoids influence cardiovascular pathology primarily through CB1 and CB2 receptors. CB1 activation exacerbates atherosclerosis by promoting inflammation, lipid accumulation, and inducing bradycardia by inhibiting norepinephrine. Conversely, CB2 activation exerts protective effects, reducing inflammatory cytokines, downregulating monocyte adhesion factors, and inhibiting NLRP3 inflammasome activation. Additionally, CB2 agonism enhances endothelial function and prevents oxidative stress. It also favors reverse cholesterol transport and stabilizes atherosclerotic plaques by reducing MMP-9 levels and collagen degradation, ultimately mitigating atherogenesis [79,81].

Some compounds modulated by orange juice intake participate in endocannabinoid signaling. PE, in addition to the effects previously mentioned, participates in endocannabinoid signaling [82]. The compound N-docosahexaenoyl phenylalanine, which is an N-acyl amide, is similar to endocannabinoids in structure and metabolism and can regulate cannabinoid physiology or operate in parallel via overlapping signaling pathways [25]. N-acyl amides are considered “orphan lipids”, since no specific receptor has been discovered for them. However, there is evidence of their ability to bind to different endocannabinoid receptors, exerting a cannabimimetic effect [83]. Activation of these receptors plays a central role in the regulation of cardiac function in both health and disease [84,85] (Figure 1).

An important relationship has been suggested between the endocannabinoid system and PPARs, especially PPARα and PPARγ. PPARs are activated by a large number of endogenous and exogenous lipid molecules, and also by endocannabinoids and endocannabinoid-like compounds. Furthermore, PPARs are indirectly modulated by receptors and enzymes that regulate the activity and metabolism of endocannabinoids, and vice versa, the expression of these receptors and enzymes may be regulated by PPARs. There is fascinating evidence that perturbation of the symbiotic gut microbial community may greatly impact on host health, either directly or indirectly, by regulating PPAR activity through endogenous, endocannabinoid-like modulators [80] (Figure 1).

### 3.3. Modulation of Gut Microbiota Diversity and Metabolism

An increasing interest has been given to the study of gut microbiota and metabolism and their influence on health and disease. Recent studies suggest a strong link between gut microbiota and development of human diseases, including obesity, insulin resistance, type 2 diabetes mellitus, and CVD [86] (Figure 1).

Metabolomic studies on orange juice intake have identified some serum metabolites derived from host gut microbiota [10,16,18,20]. Fujimori et al. [20] found changes in N2-acetyl-L-ornithine and N-formylmaleamic acid, which are metabolites from L-ornithine de novo biosynthesis and NAD biosynthesis, respectively. Kurilshikov et al. [86] studied the relationship between gut microbiome metabolism and CVD risk in healthy and obese individuals. Interestingly, L-ornithine de novo biosynthesis and NAD biosynthesis were among the metabolic pathways associated with increased CVD risk.

An experimental study in healthy rats showed that orange juice consumption improved alpha diversity and decreased the gut microbiota *Firmicutes*/*Bacteroidota* ratio (F/B ratio), preventing insulin resistance [16]. Markers of insulin resistance, poor control of blood glucose levels, and systemic inflammation were associated with lower gut microbiome diversity. Increased markers of insulin resistance, poor control of blood glucose levels, and systemic inflammation were associated with lower gut microbiome diversity. The higher alpha diversity in the orange juice intake group could reduce insulin resistance. Additionally, orange juice prevented metabolic syndrome. Furthermore, the low F/B ratio mainly comes from the high abundance of bacterial taxa from *Bacteroides*, which is negatively correlated with metabolic syndrome.

A study in healthy Brazilian volunteers investigated the impact of daily consumption of orange juices from the Cara Cara and Bahia (*Citrus sinensis* Osbeck) cultivars on the gut microbiota and metabolome. The results showed that both juices induced significant changes in microbiota composition and in the metabolites produced by the microbiota in the cecum. The major shift observed in microbiota composition was the increased abundance of a network of Clostridia OTUs. Clostridia OTUs are potentially involved in several host physiological processes, including glucose and protein metabolism, immune homeostasis, and body energy balance [10].

Cara Cara juice consumption increased the presence of *Lachnospiraceae* and *Ruminococcaceae*. Members of *Lachnospiraceae* and *Ruminococcaceae* share the capacity to generate short-chain fatty acids (SCFAs), mainly butyrate, through the fermentation of non-digestible plant fibers. Interestingly, after Cara Cara juice intake, consistent positive correlations were observed between the *Lachnospiraceae* family and the most abundant SCFAs present in the colon, including acetate, butyrate, and propionate. Cara Cara juice also increased *Mogibacteriaceae*, which appears to be associated with lower obesity levels and showed a positive correlation with isovalerate and isobutyrate, which are branched-chain fatty acids derived from leucine and valine, respectively, and they may influence systemic metabolism. Finally, Cara Cara juice increased *Parabacteroides* and *Bacteroides ovatus*, which help reduce intestinal inflammation [10].

Bahia juice increased *Adlercreutzia*, which is related to the conversion of isoflavones into equol. Equol exhibits high antioxidant activity and hormone-like activity and can act as either an agonist or antagonist of estrogen receptors. Therefore, it has promising applications in the prevention of chronic diseases such as cardiovascular disease, breast cancer, and prostate cancer. Additionally, Bahia juice increased *Veillonellaceae*, which convert lactate into propionate, another SCFA [87].

Some anaerobic gut microbes have the potential to convert dietary carbohydrates and phenolic compounds into organic acids, including SCFAs. The increase in SCFAs can also be attributed to the carbon sources present in orange juice, as carbon serves as a substrate for the fermentation of intestinal bacteria and effectively participates in SCFA production in the colon [88].

SCFAs are recognized as essential for gut physiology and host health because they function as a primary energy source for enterocytes, stimulate epithelial cell proliferation, improve blood flow, increase sodium and water absorption, decrease intraluminal pH, and ultimately reduce ammonia absorption [88]. SCFAs might also enter the systemic circulation and directly affect metabolism or the function of peripheral tissues. Increasing evidence supports a beneficial role for SCFAs in adipose tissue, skeletal muscle, and liver substrate metabolism and function, thereby contributing to improved glucose homeostasis and insulin sensitivity [89]. In fact, acetate, propionate, and butyrate can regulate hepatic lipid and glucose homeostasis by involving peroxisome proliferator-activated receptor-γ-regulated effects on gluconeogenesis and lipogenesis [90] (Figure 1).

### 3.4. Amino Acids

Valine, leucine, and isoleucine (branched-chain amino acids—BCAAs) were found to be decreased after orange juice intake in subjects with cardiovascular risk and pre- or stage 1 hypertension [6]. Numerous gut bacteria possess metabolic pathways allowing for BCAA biosynthesis. Additionally, colonic bacteria can utilize the amino acids leucine, valine, and isoleucine to generate a complex mixture of metabolic end products, including SCFAs and branched-chain fatty acids (isovalerate, isobutyrate, and isovalerate) [10,91]. Thus, the composition and metabolic performance of the intestinal microbiota contribute to BCAA availability to the host and are, at least partially, responsible for their effect on host metabolism [91] (Figure 1).

At the cellular level, BCAAs serve as direct building blocks or nitrogen donors for proteosynthesis, as an energy/anaplerotic substrate while being degraded down to the final glucogenic (propionyl-CoA and succinyl-CoA) and ketogenic (acetyl-CoA and acetoacetate) products and oxidized, or as nutritional signals via rapamycin complex (mTORC) activation [91]. BCAAs, particularly leucine, activates the mammalian target of mTORC1, the key intersection of amino acid and insulin signaling pathways, but chronic activation can inhibit the insulin receptor pathway and reduce insulin sensitivity, promoting the development of insulin resistance, obesity, and type 2 diabetes [92,93].

Additionally, mTORC1 stimulates lipogenesis, increasing fat storage in the liver and adipose tissue, which worsens dyslipidemia and hepatic steatosis [94]. mTORC1 also influences mitochondrial biogenesis and energy metabolism, but its chronic activation can lead to excessive production of reactive oxygen species and activate inflammatory factors, promoting chronic states of oxidative stress and inflammation, which are common in cardiovascular diseases [94,95]. In fact, high concentrations of BCAAs are associated with several cardiometabolic risk factors, such as insulin resistance [96], obesity, atherogenic dyslipidemia, elevated blood pressure [97,98], and an increased incidence of CVD events [99]. Thus, the decrease in these amino acids after orange juice intake may be beneficial to the heart and important in the process of cardiac remodeling.

## 4. Conclusions

Orange juice intake has shown promising effects on modulating key metabolic pathways, including phospholipid metabolism, energy regulation, endocannabinoid signaling, amino acid levels, and gut microbiota composition. The changes contribute to improved insulin sensitivity, lipid metabolism, and reduced inflammation, all of which play crucial roles in preventing or alleviating cardiac remodeling. The findings suggest that orange juice can act as a metabolic modulator, with potential therapeutic implications for CVDs. Nonetheless, further studies are needed to evaluate the metabolome in CVDs to confirm modulation of metabolic pathways by orange juice.

## Figures and Tables

**Figure 1 metabolites-15-00198-f001:**
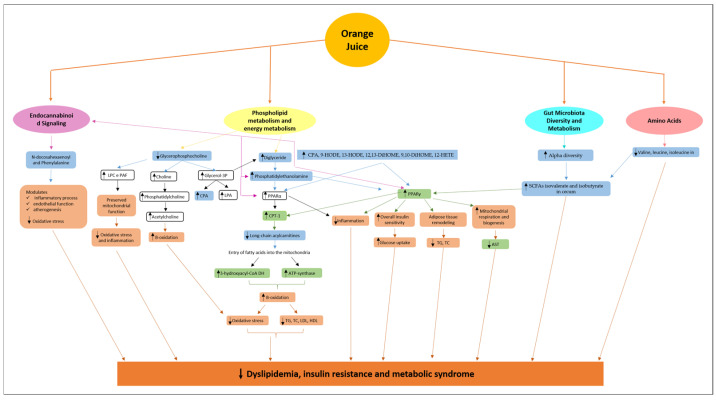
Schematic representation of action mechanisms of Orange juice. Blue box: compounds identified by metabolomic analysis after ingestion of orange juice. Green box: compounds altered by ingsetion of orange juice. Orange box: metabolic processes altered by ingestion of orange juice. Black upward arrows indicate increase and black downward arrows indicate decrease. Abbreviations: LPC: lysophosphocholines; PAF: platelet-activating factors; LPA: lysophosphatidic acid; CPA: cyclic phosphatidic acid; PPARα: peroxisome proliferator-activated receptors α; CPT-1: carnitine palmitoyltransferase 1; 3-hydroxyacyl-CoA DH: 3-hydroxyacyl-CoA dehydrogenase; TG: triglyceride; TC: total cholesterol; LDL: low-density lipoproteins; HDL: high-density lipoproteins; 9-HODE and 13-HODE: Hydroxyoctadecadienoic acid; 12,13-DiHOME and 9,10-DiHOME: Dihydroxyoctadecanoic acid; 12-HETE: 12-hydroxyeicosatetraenoic acid; PPARγ: peroxisome proliferator-activated receptors γ; AST: aspartate aminotransferase; SCFAs: short-chain fatty acids.

**Table 1 metabolites-15-00198-t001:** Metabolites altered by orange juice intake analyzed by metabolomics.

Metabolites	Biochemical Class → Function	Reference
Acyl-carnitines	Acyl group (derived from fatty acids or related compounds) + carnitine → fatty acid β-oxidation	Moreira et al. [19]
Hydroxyoctadecadienoic acid (9-HODE + 13-HODE)	Oxygenated metabolites of polyunsaturated fatty acids (linoleic acid) → PPAR ligands	Rangel-Huerta et al. [18]
Dihydroxyoctadecanoic acid (12,13-DiHOME and 9,10-DiHOME)	Oxygenated metabolites of polyunsaturated fatty acids (linoleic acid) → PPAR ligands	
12-hydroxyeicosatetraenoic acid (12-HETE)	Oxygenated metabolites of polyunsaturated fatty acids (arachidonic acid) → PPAR ligands	
N-docosahexaenoyl-phenylalanine	N-acylamides (fatty acid—acyl group- attached to a simple amine) → participate in endocannabinoid signaling	Fujimori et al. [20]
Diglyceride (DG, 20:4/24:1)	Glycerolipids → synthesis pathways of the main phospholipids and triacylglycerols in eukaryotes	
Phosphatidylethanolamine (PE, O-20:0/16:0)	Glycerophospholipid → PPAR ligands and participate in endocannabinoid signaling	
Casegravol isovalerate	Coumarins → possibly derived from oranges	
Abscisic alcohol 11-glucoside	Terpene glycosides → possibly derived from oranges	
Torvoside C	Steroidal saponins → possibly derived from oranges	
N-formylmaleamic acid	metabolites of oral and intestinal microflora → precursor in NAD synthesis	
N2-acetyl-L-ornithine	Metabolites of oral and intestinal microflora → de novo ornithine biosynthesis pathway	
Cyclic phosphatidic acid (CPA, 18:2)	Glycerophospholipid → PPAR ligands	
Proline betaine	Direct marker of citrus fruit intake	Pla-Pagà et al. [6]
Glycerophosphocholine	Small phosphodiester → compound derived from phosphatidylcholine (phospholipid) metabolism	
Acetate, valine, isoleucine, leucine	Branched-chain amino acids	
N-acetyl glycoproteins	Novel biomarker of systemic inflammation and cardiovascular disease risk	

PPAR: peroxisome proliferator-activated receptors; NAD: nicotinamide adenine dinucleotide.

## Data Availability

Not applicable.

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
