# Peer review of "The Therapeutic Potential of Orange Juice in Cardiac Remodeling: A Metabolomics Approach"

_metabolites, 2025, doi:10.3390/metabo15030198_

Round 1
Reviewer 1 Report
Comments and Suggestions for Authors
The article conveys its message with clarity and scientific soundness, and is generally acceptable. However, one important issue must be discussed:
Sometimes, during the text, the information is miscleared or difficult to comprehend. Incorporating a visual aid, such as a table or figure, could significantly enhance its readability. The organisation of ideas feels somewhat sparse at times, and readers may lose focus on the central narrative. Improved structure and the addition of visual elements would strengthen the overall presentation and impact of the article.
In addition, it would be interesting to include the search terms, databases, and any inclusion/exclusion criteria you have selected in order to carry out this review, since as it stands right now it is not clear how current the information shared is.
Comments on the Quality of English LanguageAs I said before, the paper content is interesting and complete, but the English language used occasionally poses a barrier to full comprehension (both ortography and grammar), as some phrasing is difficult to follow. For instance:
-Line 11: "Cardiovascular diseases ARE"
-Line 14: "BUT".
- Line 16: "is THE inclusion" & "A study carried...".
- Line 40: The use of "despite" is inconvenient, the sentence lacks sense.
- Line 42: "is THE inclusion" & "having" ??
- Line 43: Be careful with that citation out of the brackets.
- Line 76. "Both of..." Those two sentences should be together, to split it into two different sentences makes it meaningless.
- Be careful with the use of past-present-future tenses; try to be consequent all the time.
- The use of semicolons is way too excessive. You should remodel the way you constructed most of the sentences.
Author Response
Manuscript ID: metabolites-3372622
Opinion: Therapeutic potential of orange juice in cardiac remodeling: a metabolomics approach
Comments and Suggestions for Authors
Reviewer's Responses to Questions (red)
Dear Reviewer,
I would like to sincerely thank you for your valuable suggestions and insightful comments on my manuscript. Your feedback was extremely important in improving the quality of the article, contributing to a clearer and more rigorous presentation of the data and discussions.
I have carefully addressed all the recommendations and incorporated the necessary changes to strengthen the argument and improve the manuscript’s structure. The revisions made are highlighted in red for easier identification. I believe that, with these modifications, the article has become more robust and aligned with the expectations of the scientific community.
Once again, I truly appreciate your time and dedication in reviewing my work.
The article conveys its message with clarity and scientific soundness, and is generally acceptable. However, one important issue must be discussed:
Sometimes, during the text, the information is miscleared or difficult to comprehend. Incorporating a visual aid, such as a table or figure, could significantly enhance its readability. The organisation of ideas feels somewhat sparse at times, and readers may lose focus on the central narrative. Improved structure and the addition of visual elements would strengthen the overall presentation and impact of the article.
Thank you for your insightful feedback. Based on your suggestions, I have revised the text to improve clarity and readability. Additionally, I have incorporated a table (Table 1) and a figure (Figure 1)to better illustrate key points and enhance the overall comprehension of the manuscript. I believe these changes help to strengthen the structure and ensure a more cohesive presentation of the information.
I truly appreciate your valuable comments, which have contributed to improving the quality of the article.
In addition, it would be interesting to include the search terms, databases, and any inclusion/exclusion criteria you have selected in order to carry out this review, since as it stands right now it is not clear how current the information shared is.
Thank you for your suggestion. Initially, this opinion focused primarily on the manuscript by Fujimori et al. [20]. However, I have now included additional studies that analyzed the metabolomic effects of orange juice consumption to provide a broader perspective. Since this is an opinion article rather than a systematic review, a structured search methodology was not performed.
I have attempted to include references on orange juice consumption and metabolomic analysis to compare the altered serum metabolites in the host and their associated metabolic pathways. These are cited in section 2 and discussed in section 3 (3.1, 3.2, 3.3, and 3.4), corresponding to references 6, 16, 18, 19, and 20. Additionally, I have included studies on orange juice intake in cardiac pathological models and in patients with increased cardiovascular risk to further support and exemplify the metabolic pathways suggested as the mechanisms of action of orange juice in cardiac remodeling (lines 54 to 58 and discussed in section 3 of the manuscript, corresponding to references 4, 5, 6, 7, and 8).
I have chosen to include only studies that investigated the intake of whole orange juice and excluded studies using orange juice-derived compounds such as hesperidin and naringin. This decision was made to ensure consistency in the discussion of the metabolic effects specifically associated with orange juice consumption, as well as to emphasize the importance of the synergistic effect of the bioactive compounds present in the fruit.
Although this opinion article does not have a methods section, if the reviewer prefers, if the reviewers find it necessary, I would be happy to include a brief description of how these additional articles were selected. Please let me know if you would like me to make this adjustment.
- Buscemi, S.; Rosafio, G.; Arcoleo, G.; Mattina, A.; Canino, B.; Montana, M.; Verga, S.; & Rini, G. Effects of red orange juice intake on endothelial function and inflammatory markers in adult subjects with increased cardiovascular risk. The American journal of clinical nutrition 2012 95(5), 1089–1095. https://doi.org/10.3945/ajcn.111.031088
- Oliveira, B. C.; Santos, P. P.; Figueiredo, A. M.; Rafacho, B. P. M.; Ishikawa, L.; Zanati, S. G.; Fernandes, A. A. H.; Azevedo, P. S.; Polegato, B. F.; Zornoff, L. A. M.; Minicucci, M. F.; & Paiva, S. A. R. Influence of Consumption of Orange Juice (Citrus si-nensis) on Cardiac Remodeling of Rats Submitted to Myocardial Infarction. Arquivos Brasileiros de Cardiologia 2021, 116(6), 1127–1136. https://doi.org/10.36660/abc.20190397
- Pla-Pagà, L.; Pedret, A.; Valls, R. M.; Calderón-Pérez, L.; Llauradó, E.; Companys, J.; Martín-Luján, F.; Moragas, A.; Canela, N.; Puiggròs, F.; Caimari, A.; Del Bas, J. M.; Arola, L.; Solà, R.; & Mayneris-Perxachs, J. Effects of Hesperidin Consumption on the Cardiovascular System in Pre- and Stage 1 Hypertensive Subjects: Targeted and Non-Targeted Metabolomic Approaches (CITRUS Study). Molecular Nutrition & Food Research 2021, 65(17), e2001175. https://doi.org/10.1002/mnfr.202001175
- Ribeiro, A. P. D.; Pereira, A. G.; Todo, M. C.; Fujimori, A. S. S.; Dos Santos, P. P.; Dantas, D.; Fernandes, A. A.; Zanati, S. G.; Hassimotto, N. M. A.; Zornoff, L. A. M.; Azevedo, P. S.; Minicucci, M. F.; Paiva, S. A. R.; & Polegato, B. F. Pera Orange (Citrus sinensis) and Moro Orange (Citrus sinensis (L.) Osbeck) Juices Attenuate Left Ventricular Dysfunction and Oxidative Stress and Improve Myocardial Energy Metabolism in Acute Doxorubicin-Induced Cardiotoxicity in Rats. Nutrition (Burbank, Los Angeles County, Calif.) 2021, 91-92, 111350. https://doi.org/10.1016/j.nut.2021.1113508
- Cabral, R. P.; Ribeiro, A. P. D.; Monte, M. G.; Fujimori, A. S. S.; Tonon, C. R.; Ferreira, N. F.; Zanatti, S. G.; Minicucci, M. F.; Zornoff, L. A. M.; Paiva, S. A. R.; & Polegato, B. F. Pera Orange Juice (Citrus sinensis L. Osbeck) Alters Lipid Metabolism and Attenuates Oxidative Stress in the Heart and Liver of Rats Treated with Doxorubicin. Heliyon 2024, 10(17), e36834. https://doi.org/10.1016/j.heliyon.2024.e36834
- Wang, K.; Zhao, Y.; Xu, L.; Liao, X.; & Xu, Z. Health outcomes of 100% orange juice and orange flavored beverage: A compa-rative analysis of gut microbiota and metabolomics in rats. Current Research in Food Science 2023, 6, 100454. https://doi.org/10.1016/j.crfs.2023.100454
- Rangel-Huerta, O. D.; Aguilera, C. M.; Perez-de-la-Cruz, A.; Vallejo, F.; Tomas-Barberan, F.; Gil, A.; & Mesa, M. D. A serum metabolomics-driven approach predicts orange juice consumption and its impact on oxidative stress and inflammation in subjects from the BIONAOS study. Molecular Nutrition & Food Research 2017, 61(2), 10.1002/mnfr.201600120. https://doi.org/10.1002/mnfr.201600120
- Moreira, V.; Brasili, E.; Fiamoncini, J.; Marini, F.; Miccheli, A.; Daniel, H.; Lee, J. J. H.; Hassimotto, N. M. A.; & Lajolo, F. M. Orange juice affects acylcarnitine metabolism in healthy volunteers as revealed by a mass-spectrometry based metabolomics approach. Food Research International (Ottawa, Ont.) 2018, 107, 346–352. https://doi.org/10.1016/j.foodres.2018.02.046
- Fujimori, A.S.S., Ribeiro, A.P.D., Pereira, A.G.; Dias-Audibert, F.L.; Tonon, C.R.; Dos Santos, P.P.; Dantas, D.; Zanati, S.G.; Catharino, R.R.; Zornoff, L.A.M.; Azevedo, P.S.; de Paiva, S.A.R.; Okoshi, M.P.; Lima, E.O.; & Polegato, B.F. Effects of Pera Or-ange Juice and Moro Orange Juice in Healthy Rats: A Metabolomic Approach. Metabolites 2023, 13(8), 902. https://doi.org/10.3390/metabo13080902
Comments on the Quality of English Language
As I said before, the paper content is interesting and complete, but the English language used occasionally poses a barrier to full comprehension (both ortography and grammar), as some phrasing is difficult to follow. For instance:
-Line 11: "Cardiovascular diseases ARE"
-Line 14: "BUT".
- Line 16: "is THE inclusion" & "A study carried...".
- Line 40: The use of "despite" is inconvenient, the sentence lacks sense.
- Line 42: "is THE inclusion" & "having" ??
- Line 43: Be careful with that citation out of the brackets.
- Line 76. "Both of..." Those two sentences should be together, to split it into two different sentences makes it meaningless.
- Be careful with the use of past-present-future tenses; try to be consequent all the time.
- The use of semicolons is way too excessive. You should remodel the way you constructed most of the sentences.
Thank you for your detailed feedback on the language and clarity of the manuscript. I have now revised the English with the assistance of an expert to improve readability and correctness. The specific sentences you pointed out have been corrected, ensuring better coherence and grammatical accuracy.
Additionally, I acknowledge the excessive use of semicolons, which was an oversight. This issue has now been addressed, and the sentence structure has been improved accordingly.

Reviewer 2 Report
Comments and Suggestions for Authors
-
This paper investigates the therapeutic potential of Pera and Moro orange juices in modulating cardiac remodeling through a metabolomics approach. The study highlights changes in metabolites associated with phospholipid metabolism, the endocannabinoid system, and microbiota metabolism, providing new insights into the cardioprotective properties of these citrus juices. While the findings are promising, they are based on studies in healthy rats, and further research in pathological models is needed to validate the therapeutic effects.
1. Introduction
The introduction provides a good foundation by linking cardiac remodeling to cardiovascular diseases and highlighting the need for adjuvant therapies. However, additional references on metabolomics in cardiovascular health and prior studies using citrus-derived compounds in cardiac remodeling would enrich the background.
Consider elaborating on why Pera and Moro orange juices were chosen, including specific compounds known to have potential therapeutic effects.
2. Research Design
The research design is generally appropriate, but the inclusion of disease models in future studies would enhance the relevance of the findings.
The rationale for selecting the metabolites and pathways analyzed should be explained more thoroughly, particularly how these were prioritized for investigation.
3. Methods
Provide more detail on the metabolomics workflow, including Sample preparation steps and storage conditions; Analytical techniques (e.g., LC-MS/MS parameters); Quality control measures and validation of metabolite identification; The administration protocol for the orange juices, including dosage, duration, and frequency, should be described clearly to enable replication; Specify the statistical methods used to analyze metabolomics data and determine significance.
4. Results
The results are intriguing but would benefit from: Additional figures or tables summarizing key metabolite changes and associated pathways; A clearer presentation of how changes in specific metabolites link to the proposed mechanisms of cardiac remodeling modulation; A discussion of variability between samples, if applicable.
5. Discussion
Expand the discussion on the implications of phospholipid metabolism, endocannabinoid signaling, and microbiota changes in cardiac remodeling. Highlight how these mechanisms align with current understanding in cardiovascular research; Address limitations of the study, including the use of healthy rats instead of disease models and the need for longer-term studies to evaluate sustained effects; Suggest future research directions, such as testing in disease models or identifying specific active compounds in the juices.
6. Conclusions
The conclusions are supported by the data, but adding a brief mention of limitations and emphasizing the translational potential of the findings would strengthen the section.
- Revise grammar and sentence structure to improve readability. Specific issues include:
- Awkward phrasing (e.g., “btu it is important” should be “but it is important”).
- Repeated use of similar phrases (e.g., “almost unavoidably ends in progressive muscle dysfunction”).
- Inconsistent verb tense and article usage.
- Consider a professional language editing service for refinement.
Author Response
Manuscript ID: metabolites-3372622
Opinion: Therapeutic potential of orange juice in cardiac remodeling: a metabolomics approach
Comments and Suggestions for Authors
Reviewer's Responses to Questions (red)
Dear Reviewer,
I would like to sincerely thank you for your valuable suggestions and insightful comments on my manuscript. Your feedback was extremely important in improving the quality of the article, contributing to a clearer and more rigorous presentation of the data and discussions.
I have carefully addressed all the recommendations and incorporated the necessary changes to strengthen the argument and improve the manuscript’s structure. The revisions made are highlighted in red for easier identification. I believe that, with these modifications, the article has become more robust and aligned with the expectations of the scientific community.
Once again, I truly appreciate your time and dedication in reviewing my work.
This paper investigates the therapeutic potential of Pera and Moro orange juices in modulating cardiac remodeling through a metabolomics approach. The study highlights changes in metabolites associated with phospholipid metabolism, the endocannabinoid system, and microbiota metabolism, providing new insights into the cardioprotective properties of these citrus juices. While the findings are promising, they are based on studies in healthy rats, and further research in pathological models is needed to validate the therapeutic effects.
Thank you for your valuable feedback. In response to your comment, I have now included studies that investigated orange juice intake in pathological cardiac models and in patients with increased cardiovascular risk. These additions provide a broader perspective on the potential cardioprotective effects of orange juice beyond healthy models.
1. Introduction
The introduction provides a good foundation by linking cardiac remodeling to cardiovascular diseases and highlighting the need for adjuvant therapies. However, additional references on metabolomics in cardiovascular health and prior studies using citrus-derived compounds in cardiac remodeling would enrich the background.
Thank you for your insightful suggestion. In response, I have added additional references on metabolomics in cardiovascular health and prior studies using Orange juice in cardiac remodeling to enrich the background.
I have chosen to include only studies that investigated the intake of whole orange juice and excluded studies using orange juice-derived compounds such as hesperidin and naringin. This decision was made to ensure consistency in the discussion of the metabolic effects specifically associated with orange juice consumption, as well as to emphasize the importance of the synergistic effect of the bioactive compounds present in the fruit.
The studies investigating orange juice intake in pathological cardiac models and in patients with increased cardiovascular risk are cited in lines 54 to 58 and discussed in section 3 of the manuscript, corresponding to references 4, 5, 6, 7, and 8.
Furthermore, additional references on metabolomics after orange juice intake have been incorporated into the manuscript. These are cited in section 2 and discussed in section 3, corresponding to references 6, 16, 18, 19, and 20.
4. Buscemi, S.; Rosafio, G.; Arcoleo, G.; Mattina, A.; Canino, B.; Montana, M.; Verga, S.; & Rini, G. Effects of red orange juice intake on endothelial function and inflammatory markers in adult subjects with increased cardiovascular risk. The American journal of clinical nutrition 2012 95(5), 1089–1095. https://doi.org/10.3945/ajcn.111.031088
5. Oliveira, B. C.; Santos, P. P.; Figueiredo, A. M.; Rafacho, B. P. M.; Ishikawa, L.; Zanati, S. G.; Fernandes, A. A. H.; Azevedo, P. S.; Polegato, B. F.; Zornoff, L. A. M.; Minicucci, M. F.; & Paiva, S. A. R. Influence of Consumption of Orange Juice (Citrus si-nensis) on Cardiac Remodeling of Rats Submitted to Myocardial Infarction. Arquivos Brasileiros de Cardiologia 2021, 116(6), 1127–1136. https://doi.org/10.36660/abc.20190397
6. Pla-Pagà, L.; Pedret, A.; Valls, R. M.; Calderón-Pérez, L.; Llauradó, E.; Companys, J.; Martín-Luján, F.; Moragas, A.; Canela, N.; Puiggròs, F.; Caimari, A.; Del Bas, J. M.; Arola, L.; Solà, R.; & Mayneris-Perxachs, J. Effects of Hesperidin Consumption on the Cardiovascular System in Pre- and Stage 1 Hypertensive Subjects: Targeted and Non-Targeted Metabolomic Approaches (CITRUS Study). Molecular Nutrition & Food Research 2021, 65(17), e2001175. https://doi.org/10.1002/mnfr.202001175
7. Ribeiro, A. P. D.; Pereira, A. G.; Todo, M. C.; Fujimori, A. S. S.; Dos Santos, P. P.; Dantas, D.; Fernandes, A. A.; Zanati, S. G.; Hassimotto, N. M. A.; Zornoff, L. A. M.; Azevedo, P. S.; Minicucci, M. F.; Paiva, S. A. R.; & Polegato, B. F. Pera Orange (Citrus sinensis) and Moro Orange (Citrus sinensis (L.) Osbeck) Juices Attenuate Left Ventricular Dysfunction and Oxidative Stress and Improve Myocardial Energy Metabolism in Acute Doxorubicin-Induced Cardiotoxicity in Rats. Nutrition (Burbank, Los Angeles County, Calif.) 2021, 91-92, 111350. https://doi.org/10.1016/j.nut.2021.1113508
8. Cabral, R. P.; Ribeiro, A. P. D.; Monte, M. G.; Fujimori, A. S. S.; Tonon, C. R.; Ferreira, N. F.; Zanatti, S. G.; Minicucci, M. F.; Zornoff, L. A. M.; Paiva, S. A. R.; & Polegato, B. F. Pera Orange Juice (Citrus sinensis L. Osbeck) Alters Lipid Metabolism and Attenuates Oxidative Stress in the Heart and Liver of Rats Treated with Doxorubicin. Heliyon 2024, 10(17), e36834. https://doi.org/10.1016/j.heliyon.2024.e36834
16. Wang, K.; Zhao, Y.; Xu, L.; Liao, X.; & Xu, Z. Health outcomes of 100% orange juice and orange flavored beverage: A compa-rative analysis of gut microbiota and metabolomics in rats. Current Research in Food Science 2023, 6, 100454. https://doi.org/10.1016/j.crfs.2023.100454
18. Rangel-Huerta, O. D.; Aguilera, C. M.; Perez-de-la-Cruz, A.; Vallejo, F.; Tomas-Barberan, F.; Gil, A.; & Mesa, M. D. A serum metabolomics-driven approach predicts orange juice consumption and its impact on oxidative stress and inflammation in subjects from the BIONAOS study. Molecular Nutrition & Food Research 2017, 61(2), 10.1002/mnfr.201600120. https://doi.org/10.1002/mnfr.201600120
19. Moreira, V.; Brasili, E.; Fiamoncini, J.; Marini, F.; Miccheli, A.; Daniel, H.; Lee, J. J. H.; Hassimotto, N. M. A.; & Lajolo, F. M. Orange juice affects acylcarnitine metabolism in healthy volunteers as revealed by a mass-spectrometry based metabolomics approach. Food Research International (Ottawa, Ont.) 2018, 107, 346–352. https://doi.org/10.1016/j.foodres.2018.02.046
20. Fujimori, A.S.S., Ribeiro, A.P.D., Pereira, A.G.; Dias-Audibert, F.L.; Tonon, C.R.; Dos Santos, P.P.; Dantas, D.; Zanati, S.G.; Catharino, R.R.; Zornoff, L.A.M.; Azevedo, P.S.; de Paiva, S.A.R.; Okoshi, M.P.; Lima, E.O.; & Polegato, B.F. Effects of Pera Or-ange Juice and Moro Orange Juice in Healthy Rats: A Metabolomic Approach. Metabolites 2023, 13(8), 902. https://doi.org/10.3390/metabo13080902
Consider elaborating on why Pera and Moro orange juices were chosen, including specific compounds known to have potential therapeutic effects.
Thank you for your suggestion. I believe this discussion no longer needs to be added to the manuscript, as it was originally specific to the study by Fujimori et al. [20], which was the sole study referenced in the previous version of the opinion article. With the inclusion of new studies, other types of oranges have also been cited, making the focus on Pera and Moro juices less relevant to the overall discussion.
20. Fujimori, A.S.S., Ribeiro, A.P.D., Pereira, A.G.; Dias-Audibert, F.L.; Tonon, C.R.; Dos Santos, P.P.; Dantas, D.; Zanati, S.G.; Catharino, R.R.; Zornoff, L.A.M.; Azevedo, P.S.; de Paiva, S.A.R.; Okoshi, M.P.; Lima, E.O.; & Polegato, B.F. Effects of Pera Or-ange Juice and Moro Orange Juice in Healthy Rats: A Metabolomic Approach. Metabolites 2023, 13(8), 902. https://doi.org/10.3390/metabo13080902
2. Research Design
The research design is generally appropriate, but the inclusion of disease models in future studies would enhance the relevance of the findings.
The studies investigating orange juice intake in pathological cardiac models and in patients with increased cardiovascular risk are cited in lines 54 to 58 and discussed in section 3 of the manuscript, corresponding to references 4, 5, 6, 7, and 8.
Furthermore, additional references on metabolomics after orange juice intake have been incorporated into the manuscript. These are cited in section 2 and discussed in section 3, corresponding to references 6, 16, 18, 19, and 20.
The rationale for selecting the metabolites and pathways analyzed should be explained more thoroughly, particularly how these were prioritized for investigation.
I have aimed to include all host serum metabolites that were found to be altered after orange juice intake in metabolomics studies. I did not add this information explicitly in the text, as this is an opinion article and does not have a methodology section. However, if the reviewer finds it necessary, I can find an appropriate place to include it. The metabolites identified are listed in Table 1.
3. Methods
Provide more detail on the metabolomics workflow, including Sample preparation steps and storage conditions; Analytical techniques (e.g., LC-MS/MS parameters); Quality control measures and validation of metabolite identification; The administration protocol for the orange juices, including dosage, duration, and frequency, should be described clearly to enable replication; Specify the statistical methods used to analyze metabolomics data and determine significance.
Thank you for your insightful comments. I would like to clarify that this is an opinion article, not a research study with a detailed methodology section. In this revised version, several new studies on orange juice intake and metabolomic analysis have been included to enrich the discussion. As a result, each study presents different methodologies and study designs. However, if the reviewer finds it necessary, I can provide a summary table detailing the key methodological aspects of each metabolomic study included.
4. Results
The results are intriguing but would benefit from: Additional figures or tables summarizing key metabolite changes and associated pathways; A clearer presentation of how changes in specific metabolites link to the proposed mechanisms of cardiac remodeling modulation; A discussion of variability between samples, if applicable.
An additional figure (Figure 1) and table (Table 1) summarizing key metabolite changes and associated pathways have been added to the manuscript. Figure 1 aims to provide a clearer presentation of how changes in specific metabolites are linked to the proposed mechanisms of cardiac remodeling modulation.
5. Discussion
Expand the discussion on the implications of phospholipid metabolism, endocannabinoid signaling, and microbiota changes in cardiac remodeling. Highlight how these mechanisms align with current understanding in cardiovascular research; Address limitations of the study, including the use of healthy rats instead of disease models and the need for longer-term studies to evaluate sustained effects; Suggest future research directions, such as testing in disease models or identifying specific active compounds in the juices.
The discussion on the implications of phospholipid metabolism, endocannabinoid signaling, and microbiota changes in cardiac remodeling has been expanded in Section 3 (3.1, 3.2, 3.3, and 3.4) of the manuscript. Additionally, studies on orange juice consumption in cardiac pathological models and individuals with increased cardiovascular risk have been included. However, it has been emphasized that further studies are needed to evaluate the metabolome in CVDs to confirm the modulation of metabolic pathways by orange juice.
6. Conclusions
The conclusions are supported by the data, but adding a brief mention of limitations and emphasizing the translational potential of the findings would strengthen the section.
Comments on the Quality of English Language
· Revise grammar and sentence structure to improve readability. Specific issues include:
· Awkward phrasing (e.g., “btu it is important” should be “but it is important”).
· Repeated use of similar phrases (e.g., “almost unavoidably ends in progressive muscle dysfunction”).
· Inconsistent verb tense and article usage.
· Consider a professional language editing service for refinement.
Thank you for your detailed feedback on the language and clarity of the manuscript. I have now revised the English with the assistance of an expert to improve readability and correctness. The specific sentences you pointed out have been corrected, ensuring better coherence and grammatical accuracy.

Reviewer 3 Report
Comments and Suggestions for Authors
Please read the whole manuscript carefully and change according to the comments, and suggestions.

Have some errors, need to improve
Author Response
Manuscript ID: metabolites-3372622
Opinion: Therapeutic potential of orange juice in cardiac remodeling: a metabolomics approach
Comments and Suggestions for Authors
Reviewer's Responses to Questions (red)
Dear Reviewer,
I would like to sincerely thank you for your valuable suggestions and insightful comments on my manuscript. Your feedback was extremely important in improving the quality of the article, contributing to a clearer and more rigorous presentation of the data and discussions.
I have carefully addressed all the recommendations and incorporated the necessary changes to strengthen the argument and improve the manuscript’s structure. The revisions made are highlighted in red for easier identification. I believe that, with these modifications, the article has become more robust and aligned with the expectations of the scientific community.
Once again, I truly appreciate your time and dedication in reviewing my work.
- The abstract and keywords have been revised.
- The references have been formatted
- Citrus sinensis has been italicized.
